# Iron, the Essential Micronutrient: A Comprehensive Review of Regulatory Pathways of Iron Metabolism

**DOI:** 10.3390/nu18010109

**Published:** 2025-12-28

**Authors:** Adrienn Horváth, Kitti Tamási, Ramóna Pap, Gergely Jánosa, Edina Pandur

**Affiliations:** Department of Pharmaceutical Biology, Faculty of Pharmacy, University of Pécs, H-7624 Pécs, Hungary; horvath.adrienn2@pte.hu (A.H.); tamasi.kitti@gytk.pte.hu (K.T.); pap.ramona@pte.hu (R.P.); janosa.gergely@gytk.pte.hu (G.J.)

**Keywords:** iron, regulatory pathway, hepcidin, HAMP, BMP/SMAD, JAK-STAT

## Abstract

Iron constitutes an essential micronutrient in living organisms. All iron is absorbed into the body through dietary intake, except for exogenous therapeutic sources. Dietary iron is typically categorized as either heme or nonheme iron. Nonheme iron is essential for regulating iron in the body, as it exists in various forms, including soluble iron, storage iron within ferritin, and iron found in the catalytic centers of a wide range of proteins. Iron homeostasis is carefully managed to ensure that sufficient iron is available for critical biological processes while preventing the harmful effects that can arise from excess iron. The small peptide hormone hepcidin is the main regulator of iron homeostasis. Hepcidin and other iron regulatory molecules are regulated by various signaling pathways, such as IL-6/JAK-STAT, BMP/SMAD, and MAPK. Alterations in regulatory pathways may occur in response to iron accumulation or deficiency. Iron overload in the body can activate JAK/STAT, BMP/SMAD and MAPK pathways, leading to the initiation hepcidin synthesis. Conversely, in iron deficiency, as in hypoxic conditions or EPO-mediated signaling pathways, HAMP synthesis in the nucleus is reduced. Thus, this review provides an update on the possible regulatory pathways that play a role in iron regulation and may be potential therapeutic targets.

## 1. Introduction

Iron constitutes a vital trace element in living organisms. Iron homeostasis is carefully regulated to ensure that sufficient iron is available for critical biological processes while preventing the harmful effects of excess iron [1]. Serum iron levels and transferrin saturation are commonly utilized indicators of the body’s iron status [2]. The liver, placenta, intestine, and macrophages are the primary organs and tissues involved in the regulation of iron stores [3]. The oxidation state of iron significantly influences its gastrointestinal absorption. For effective absorption by intestinal epithelial cells, iron must be in the ferrous state or associated with a transporter protein [4]. Cells can absorb and utilize iron through the action of divalent metal transporter-1 (DMT-1) and transferrin receptor-1 (TfR1) iron importers [5].

Hepcidin plays a significant role in regulating iron homeostasis. Hepcidin, a small peptide hormone secreted by the liver, is the primary regulator of iron homeostasis [6]. Hepcidin is encoded by the hepcidin antimicrobial peptide (HAMP) gene and synthesized in hepatocytes [7]. These molecules play crucial roles in the regulation of body iron stores. Hepcidin reduces the absorption of iron in the intestines and the release of iron from macrophages by attaching to the iron-exporting protein ferroportin (FPN1), which results in its internalization and eventual breakdown [8]. However, hepcidin-independent FPN1 internalization has been observed in macrophages [9]. Inflammatory conditions significantly influence hepcidin synthesis and play a pivotal role in the regulation of iron homeostasis [10]. Hepcidin expression increases in response to inflammatory conditions, potentially contributing to iron accumulation in neurons [11]. Iron may play a role in certain conditions, such as the implantation of embryos or in neurodegenerative diseases like Parkinson’s disease. Some special conditions, such as embryo implantation or neurodegenerative diseases, such as Parkinson’s disease, may depend on iron. Prior to implantation, the embryo must absorb essential nutrients, such as iron and vitamins, from the uterine cavity. Consequently, both iron deficiency and iron overload may increase the risk of infertility [12]. Numerous studies have revealed an increase in the total iron concentration within the substantia nigra of individuals diagnosed with Parkinson’s disease [13]. Iron accumulation within brain cells, facilitated by inflammation, neurodegeneration, brain injury, or infections, results in oxidative damage, mitochondrial dysfunction, and cell death [14]. Hepcidin expression is modulated by various extracellular signals, including inflammation, iron availability, hypoxia, special endocrine signals, and glucose metabolism, through distinct signal transduction pathways, such as Interleukin-6/Janus Kinase-Signal Transducer and Activator of Transcription (IL-6/JAK-STAT), Bone Morphogenetic Protein/Sma and Mad Homologs (BMP/SMAD), and Mitogen-activated protein kinase (MAPK).

## 2. Materials and Methods

### 2.1. Literature Search

This review adhered to the PRISMA (Preferred Reporting Items for Systematic Reviews and Meta-Analyses) guidelines. PubMed/Medline, Web of Science, and Scopus were searched for eligible studies. The primary search strings included: hepcidin, iron regulation, and signaling pathway.

The research was confined to English-language sources published from 1997 to 2025. Studies conducted before 2000 mainly focused on the general effects of iron and hepcidin. This timeframe was selected to guarantee both scientific significance and methodological rigor. No restrictions were imposed on the study type, permitting the inclusion of in vitro, in vivo, and clinical studies, as well as narrative or systematic or reviews. Furthermore, the reference lists of key articles were manually reviewed to discover additional eligible studies.

### 2.2. Focus Question

In what ways do iron regulatory pathways vary in response to different iron levels and the presence of disease?

### 2.3. Study Selection and Inclusion Criteria

Studies were included if they:investigated the regulatory role or cellular mechanisms of hepcidin or iron;documented preclinical findings from either in vitro or in vivo experiments or clinical proof of hepcidin’s impact;addressed bioavailability, iron overload, or iron deficiency delivery strategies context

The exclusion criteria were as follows:non-peer-reviewed publicationsstudies focused only on ferroptosis conditions;duplicate or non-English publications.

### 2.4. Screening Process

The process of selecting studies was carried out following the PRISMA 2020 guidelines, identifying a total of 560 records across three databases: PubMed (*n* = 406), Web of Science (*n* = 289), and Scopus (*n* = 271). After eliminating 252 duplicate entries, 308 distinct publications were considered suitable for the initial screening phase, during which 103 were dismissed based on their title and abstract. Consequently, 205 reports were selected for full-text review; however, 12 were excluded because they included tumor disease, leaving 193 to be evaluated for adherence to the inclusion criteria (Figure 1).

## 3. Overview of Iron Metabolism

The primary forms of iron present in the human diet include heme, ferritin, and ferric iron, which are complexed with other macromolecules [15]. Iron absorption occurs in the small intestine via DMT-1. Dietary non-heme iron predominantly exists as iron (III) (Fe^3+^) and requires reduction for its transport. This reduction is facilitated by cytochrome B (DcytB), a reductase located in the apical membrane of intestinal enterocytes. Simultaneously, iron is absorbed in the form of heme; however, the transporter responsible for heme uptake has not yet been definitively identified. Iron export via FPN is contingent upon two multicopper oxidases: ceruloplasmin (Cp) in circulation and hephaestin in the basolateral membrane of enterocytes. These oxidases facilitate the conversion of iron from Fe^2+^ to Fe^3+^ and enable its incorporation into transferrin (Tf). Within intestinal enterocytes, cytosolic iron is either sequestered in ferritin or transported into the plasma via the basolateral iron-exporting FPN [16].

Following its translocation through the membrane, facilitated by the iron exporter FPN [17], iron predominantly enters the serum as Fe^3+,^ where it binds to Tf. Iron is distributed to tissues via blood plasma, which contains only 2–4 mg of iron bound to the iron-transport protein transferrin. Plasma iron undergoes turnover every few hours, with 20–25 mg of iron per day moving through the vascular compartment [15]. Iron is subsequently transported to various cells, including those involved in heme production, iron uptake, and storage. On the cell membrane, iron-bound transferrin attaches to transferrin receptor 1 (TfR1) and is taken into the cell as the Tf (Fe3+)–TfR1 complex through receptor-mediated endocytosis (RME). Within cells, iron is distributed into three distinct compartments: the functional compartment, where mitochondria play a crucial role in preparing iron cofactors for proteins that utilize iron in metabolic activities; the storage compartment; and the labile iron pool, which contains intracellular free iron [18].

Systemic iron homeostasis is regulated by a 25-amino acid peptide, hepcidin. Hepcidin is secreted by hepatocytes in response to conditions such as iron overload, inflammation, anemia, or hypoxia [19]. It plays a role in regulating iron balance by attaching to FPN, which results in FPN being phosphorylated, then broken down, and ultimately prevents iron from being exported from cells, thus lowering the iron levels in the blood [20]. Hepcidin binds to ferroportin, blocking its channel and hindering its ability to export iron, which results in ferroportin being internalized and degraded in lysosomes. This mechanism effectively limits the amount of iron that enters the bloodstream. The transcription of hepcidin is increased in response to iron or inflammatory signals through the BMP/SMAD and IL-6/STAT3 signaling pathways, respectively [21].

Iron absorption in the human body is subject to stringent regulation. The concentration of iron in plasma is maintained within a stable range of 10–30 mmol/L, as the influx of iron into the plasma is counterbalanced by its efflux. This influx is primarily determined by the release of iron from macrophages during the recycling of senescent red blood cells, as well as dietary iron absorption and the mobilization of iron stored in the liver. The most substantial flux, approximately 20 mg/day, originates from iron-recycling macrophages, whereas dietary iron absorption accounts for only 1–2 mg/day. Under typical physiological conditions, iron is excreted through mechanisms such as blood loss, perspiration, and desquamation of epithelial cells, with these losses collectively amounting to approximately 1–2 mg of iron per day [22].

Various dietary patterns can influence the body’s iron levels. In addition to the standard diet, vegetarian and Mediterranean diets are among the most prevalent. Vegetarians who consume a diverse and well-balanced diet are not at an increased risk of iron deficiency anemia compared to non-vegetarians. To maintain sufficient iron levels, it is important to consume a diet rich in a variety of whole grains, legumes, nuts, seeds, dried fruits, iron-enriched cereals, and green leafy vegetables. The gastrointestinal tract plays a crucial role in regulating the absorption of non-heme iron, which is significantly improved by the presence of vitamin C and other organic acids [23]. The Mediterranean diet may be characterized by low iron availability, as it contains a limited amount of iron and includes inhibitors of iron absorption, such as polyphenols and phytates, which outweigh enhancers such as vitamin C or red meat. Consequently, the traditional Mediterranean diet offers the benefit of reducing the risk of severe degenerative diseases [24].

Iron deficiency (ID) denotes a reduction in total body iron stores, which may result from inadequate nutrition, diminished absorption due to gastrointestinal conditions, increased blood loss, or heightened requirements, as observed during pregnancy. Iron’s typically low bioavailability can result in either iron deficiency anemia or a non-anemic form of iron deficiency [25]. Iron deficiency anemia (IDA) is characterized by reduced hemoglobin (Hb) or hematocrit levels, accompanied by microcytic and hypochromic red blood cells (RBC), as well as a decreased red blood cell count due to insufficient iron. This condition predominantly affects women of childbearing age, developing fetuses, children, individuals with chronic inflammatory conditions, and the elderly. Iron deficiency is linked to fatigue and can lead to impairments in the immune system, growth, and neurocognitive functions. Oral iron supplements, including iron salts such as ferrous fumarate or ferrous ascorbate, are well established in clinical practice [26].

Nevertheless, iron overload can also be observed in conditions such as thalassemia. This genetic disorder affects blood-forming organs and is characterized by reduced hemoglobin production, resulting in persistent anemia. A major complication associated with thalassemia is iron overload, primarily caused by iron absorption in the gastrointestinal tract and frequent blood transfusion. This condition can eventually lead to iron-overload cardiomyopathy, which is a leading cause of illness and death among patients with thalassemia [27].

The regulation of HAMP, as can be seen, depends on many factors, not only on iron uptake or release, but also on metabolism. Notably, inflammatory factors serve as primary regulators of iron, given the rapid response of the body to these stimuli.

Imbalance in iron regulation can result in either a deficiency or an excess of iron. These conditions can be treated in several ways, depending on their severity. Iron deficiency is the most prevalent cause of anemia, and it can frequently be addressed through an iron supplement, which can often be treated with a proper diet. This diet includes red meat, fish, and poultry, which are abundant sources of heme iron. In contrast, foods that contain only non-heme iron include legumes, eggs, dark leafy greens, nuts, seeds, dried fruits, and whole grains [28]. Iron deficiency, irrespective of its cause, necessitates correction through iron supplementation, which can be administered either orally or intravenously [29].

However, the other extreme is the accumulation of excess iron in the body, which leads to the production of free radicals and can cause significant adverse effects on health [30]. To counteract the detrimental effects of iron toxicity, medical protocols have incorporated the use of iron-chelating agents [31]. Metal ions are sequestered, which significantly diminishes their reactivity. Consequently, iron-chelating agents mitigate iron accumulation in various organs, including the liver or heart [32,33]. Furthermore, the chelation therapy facilitates the regulation of iron levels accumulated from blood transfusions by enhancing the excretion of iron through urine and/or feces via chelating agents [34].

Biological therapy offers a new method for addressing diseases related to iron. Enhancing the levels of circulating hepcidin could be advantageous for individuals suffering from iron-overload conditions caused by a lack of hepcidin, such as hereditary hemochromatosis and β-thalassemia. This enhancement can be accomplished by using agents that either imitate hepcidin’s function or boost the body’s own production of hepcidin, like minihepcidins [35]. Serum hepcidin levels are elevated in anemia associated with inflammation and chronic kidney disease. Agents that reduce hepcidin production include those that antagonize hepcidin-stimulatory pathways, such as BMP or JAK/STAT signaling, as well as those that enhance suppressive pathways, including erythroid factors. Hepcidin-neutralizing agents encompass antihepcidin antibodies, such as anti-ferroportin antibodies, or engineered protein-based binders like anticalins [36].

## 4. Regulatory Pathways

### 4.1. An Overview of Regulatory Pathways

Since iron is essential for cell function, regulatory pathways work together to maintain the iron levels required by the body at all times, adapting to the body’s needs and different conditions (Table 1). The strongest stimulus for the modification of hepcidin synthesis is inflammation, which increases hepcidin production by triggering HAMP expression through the JAK/STAT pathway [37]. Consequently, serum iron levels and iron release from cells decrease due to the internalization of the FPN iron exporter [38].

The iron level in the extracellular space is monitored by the HFE/TfR2 system. When the level of transferrin-bound iron increases, HFE binds to TfR2 and initiates hepcidin transcription via the MAPK pathway [39,40]. The MAPK pathway can also be activated by the fractalkine/fractalkine receptor (CX3CR1) axis in certain cells, leading to serine phosphorylation of STAT factors and increased transcription of HAMP [11].

On the other hand, NFκB and TNF-α crosstalk on HAMP expression, as well as IL-1β-regulated C/EBPα, all increase HAMP transcription [41,42]. Moreover, the BMP/SMAD signaling pathway can be activated by both inflammation and iron availability, enhancing HAMP transcription [38]. It has been revealed that IL-6 activates both STAT3 and SMAD1/5/9, increasing HAMP expression [43].

The negative HAMP regulator HIF1α acts directly on the HAMP promoter and also downregulates HAPM expression via the inhibition of BMP/SMAD signaling pathway [44,45,46]. In addition to the interactions mentioned above, insulin, erythroferrone, growth hormones, and estrogen also contribute to HAMP regulation and thus to the maintenance of iron homeostasis via linking to the STAT and SMAD pathways [47,48,49,50].

### 4.2. Inflammation-Mediated Signaling Pathway

Inflammatory cytokines, such as interleukin-6 (IL-6) and IL-22, initiate inflammatory responses. These cytokines are necessary for hepcidin induction [51]. IL-6 and IL-22 are cytokines produced by macrophages during inflammation. This process is mediated by the JAK/STAT signaling pathway [52]. Interleukins are released during inflammation and bind to the gp130 protein receptor complex. This leads to the phosphorylation of the transcription factor STAT3, which is mediated by JAK1/2 tyrosine kinase. Upon activation, STAT3 translocates to the cell nucleus, where it binds to the hepcidin promoter, initiating hepcidin transcription (Figure 2) [37].

Another inflammatory pathway may be activated by activin B, a signaling molecule within the TGF superfamily that can bind to type II serine-threonine kinase receptors that overlap with the BMP subfamily. This interaction induces the phosphorylation of SMADs, which are activated by the receptor. The phosphorylated SMAD1/5/8 complex subsequently translocates to the nucleus in conjunction with co-SMAD4, where it regulates gene transcription (Figure 2) [38].

In inflammatory conditions, fractalkine plays a role in regulating iron metabolism by facilitating the production of proinflammatory cytokines through NFκB pathway activation [53]. NFκB is essential for regulating the expression of genes linked to inflammation. It facilitates the production of cytokines, such as IL-1β, IL-6, IL-8, and TNF-α [41]. Fractalkine can regulate cytokine production via the fractalkine receptor complex (CX3CR1) and, in certain aspects, maintain the resting state of microglia. Soluble fractalkine may facilitate the internalization of CX3CR1, potentially impacting the tyrosine phosphorylation of STAT proteins through the mitogen-activated protein kinase (MAPK) signaling pathway. Furthermore, p-STAT3 and p-STAT5 may influence HAMP transcription, potentially leading to its upregulation in microglia [11].

In hepatic inflammation, Kupffer cells respond to pro-inflammatory stimuli by accelerating IL-1β production [54]. The cytokine IL-1β induces the synthesis of CCAAT/enhancer-binding protein delta (C/EBPδ) and interleukin-6 (IL-6). C/EBPδ binds to the C/EBP binding site (C/EBP-BS) on the hepcidin promoter to initiate transcription. Simultaneously, Interleukin-6 (IL-6) induces the phosphorylation of signal transducer and activator of transcription 3 (STAT3), which subsequently enhances the transcription of hepcidin through the STAT binding site (STAT-BS) [55]. When LPS triggers IL-1β expression, it increases C/EBPδ levels, thereby enhancing the ability of C/EBPδ to bind to the C/EBP-BS on the hepcidin promoter [56].

CCAAT/enhancer-binding protein α (C/EBPα) plays a regulatory role in hepatic iron metabolism. Hepcidin, identified as a target of C/EBPα, is posited to function as a soluble modulator within this metabolic pathway. If C/EBPα expression decreases (hepatic fibrosis), it reduces hepcidin levels in hepatocytes [42]. Reduced hepcidin levels result in iron accumulation within hepatocytes, which can enhance the Fenton reaction, leading to the production of substantial reactive oxygen species (ROS) that cause significant cellular and tissue damage [57].

In human myoblasts, tumor necrosis factor alpha (TNF-α) has been shown to increase the synthesis of ferritin H (heavy) by transcriptional dependence on iron and ferritin levels. In contrast, ferritin L (light) is unaffected [58]. TNF-α induces ferritin to sequester iron, thereby adjusting the level or activity of ferritin in relation to the intracellular iron concentration. Therefore, as intracellular ferritin levels increase, a larger proportion of iron is incorporated into ferritin, regardless of the actual quantity of iron supplied to the cells [59]. Iron bound to transferrin is taken up by cells from the plasma through receptor-mediated endocytosis involving TFR1 [60]. Moreover, TNF-α enhances the expression of TFR2 while concurrently inhibiting the expression of DMT-1 and iron-regulated transporter 1 (IREG1), which are critical for cellular iron uptake and release [61].

Activated Kupffer cells are the primary sources of TNF-α in the liver. TNF-α decreases the amount of chelatable iron in hepatocytes [62]. In addition, TNF-α has other roles in hepatocytes, such as an apoptosis-inducing factor during liver diseases associated with chronic inflammation [63]. In addition, TNF-α inhibits HAMP expression in hepatoma cells [51]. However, whether this signaling is solely mediated by IL-6 remains to be elucidated.

### 4.3. Iron-Mediated Signaling Pathway

The bone morphogenic protein (BMP) signaling pathway is controlled by hepcidin transcription and transferrin-bound iron (Tf-Fe) levels in the blood. Tf-Fe consists of hemojuvelin (HJV), hemochromatosis protein (HFE), transferrin receptor 1 (TFR1), and transferrin receptor 2 (TFR2). These interactions between the three membrane proteins participate in the transmission of increased Tf-Fe levels and the activation of hepcidin transcription [64,65,66]. The HFE protein forms complexes with TFR1, facilitating the delivery of transferrin-bound iron (Fe-Tf) to a majority of cell types [67]. Additionally, Fe-Tf competes with transferrin for the same binding sites on TfR1 [68]. Intracellularly, a decrease in the amount of iron in cells stabilizes the TfR1 transcript, increases the amount of TfR1 protein, enhances HFE secretion, and reduces the hepcidin expression. Extracellularly, Fe-Tf competes effectively with hereditary HFE to bind TFR1, allowing hepcidin upregulation [69]. TfR2 is predominantly found in hepatocytes and erythroid precursors [70]. TfR2 has a low affinity for binding Fe-Tf and, unlike TfR1, interacts with both HFE and Fe-Tf simultaneously [71]. The interaction between HFE and TfR1 is inversely correlated with Fe-Tf saturation levels. When serum Fe-Tf concentrations increase, HFE is displaced from TfR1, enabling it to bind TfR2 and initiate hepcidin transcription via the MAPK pathway (Figure 3) [39,40].

Bone morphogenetic proteins (BMPs) are constituents of the transforming growth factor β (TGF-β) superfamily of cytokines. BMP ligands, including BMP6 and BMP4, interact with Type I and Type II BMP receptors [72]. There are four type I BMP receptors and three type II BMP receptors. BMP type I receptors include activin-like kinase 1 (ALK1), ALK2, ALK3, and ALK6. ALK1 is expressed in epithelial cells. ALK2 and ALK3 are also expressed in hepatocytes. ALK6 is minimally expressed in hepatocytes, with its most significant expression found in the lungs, brain, and ovaries [73,74]. BMP type II receptors include BMPR2, activin type IIA (ActRIIA), and activin type IIB (ActRIB). BMP2/7/9 binds to these receptors [75]. These receptors activate and trigger the phosphorylation of receptor-associated SMAD proteins (R-SMADs). The R-SMADs involved are SMAD1, SMAD5, and SMAD8, which subsequently bind to SMAD4. This complex then translocates to the nucleus, resulting in hepcidin transcription [76,77]. Three negative regulators of this mechanism exist: SMAD6, SMAD7, and BMPER [78]. BMP-binding endothelial cell precursor-derived regulator (BMPER) inhibits BMP2 and BMP6 binding, leading to decreased hepcidin expression (Figure 3) [79].

HJV, encoded by HFE2, is an upstream regulator of hepcidin expression [80]. The membrane-bound HJV functions as a co-receptor for both BMP2 and BMP4. It can bind to BMP2, BMP4, and BMP6 and forms a complex with BMPR [81,82]. HJV also interacts with neogenin [83], a transmembrane protein essential for the cleavage of membrane-bound HJV, to form soluble HJV. Neogenin subsequently activates transmembrane serine protease (TMPRSS6), also known as matriptase-2 [84]. Matriptase-2 inhibits BMP-induced hepcidin transcription activation by proteolytically cleaving the BMP co-receptor hemojuvelin, which is located on the cell surface [85]. This process results in a reduction in BMP-mediated hepcidin induction by inhibiting the HJV BMP coreceptor (Figure 3) [64].

### 4.4. Hypoxia-Mediated Signaling Pathway

Hypoxia can occur under various biological conditions. In such circumstances, cells adjust to low-oxygen environments by upregulating the expression of various genes, including erythropoietin (EPO), glycolytic enzymes, and genes related to iron regulation, such as transferrin receptor, transferrin, and ceruloplasmin [86,87].

Hypoxia elevates iron requirements and reduces hepcidin levels, thereby facilitating dietary iron absorption and mobilization of iron from stores [88]. Hypoxia induces an increase in EPO levels, which plays a vital role in red blood cell synthesis [89]. Hypoxia-inducible factor (HIF) directly regulates the synthesis of hepatic EPO under hypoxic conditions when erythropoiesis is pharmacologically inhibited [90].

The hypoxia induction factors include hypoxia-inducible factor 1 (HIF-1) and hypoxia-inducible factor 2 (HIF-2). Hypoxia-inducible factors (HIFs) are heterodimeric transcription factors consisting of alpha and beta subunits. HIF-1 is a transcriptional activator that relies on oxygen and is composed of a constantly expressed HIF-1β subunit along with one of three possible subunits: HIF-1α, HIF-2α, or HIF-3α [91]. The functionality of the HIF transcription factor is modulated by the stability of its α-subunits and their ability to interact with the p300 transcriptional co-activator, which is contingent on oxygen levels [92].

Under hypoxic conditions, the HIF-1α subunit plays a critical role. HIF-1α interacts with various protein factors, including prolyl-hydroxylase (PHD) and von Hippel-Lindau protein (pVHL). Under hypoxic conditions, prolyl hydroxylase domain (PHD) is inhibited, thereby allowing the stabilization of hypoxia-inducible factors HIF-1 and HIF-2 (Figure 4). These elements subsequently influence the transcription of various genes associated with iron metabolism, such as Tf, DMT-1, and FPN1 [93], and the expression levels of hepcidin mRNA are reduced [44].

Hypoxia also downregulates hepcidin expression by inhibiting the SMAD4 signaling pathway. It has been shown that SMAD4 reduction may be the central mechanism of hepcidin suppression (Figure 4) [94,95].

The indirect transcriptional regulation of hepcidin by HIF may be partially mediated by furin, which is also subject to transcriptional regulation in response to iron, oxygen, and HIF levels. HIF-1 is required for hypoxic activation of the furin promoter, thereby significantly increasing furin-encoding fur mRNA levels in response to hypoxic challenge [46]. Since furin increases the concentration of soluble HJV, it indirectly reduces hepcidin expression. Additionally, HIF-1α also enhances the transcription of TMPRSS6, which counteracts BMP signaling in liver cells by cleaving membrane-bound HJV, decreasing hepcidin expression (Figure 4) [45].

### 4.5. Endocrine Signals-Mediated Signaling Pathway

#### 4.5.1. Erythroferrone

Erythroferrone (ERFE), synthesized by erythroblasts, acts as the primary erythroid regulator of hepcidin, a homeostatic hormone that controls plasma iron levels and total body iron [96]. Following hemorrhage or erythropoietin (EPO) administration, ERFE inhibits hepcidin expression in the hepatocytes [49]. EPO production increases in response to anemia. EPO leads to a reduction in hepcidin, partly by enhancing ERFE production. ERFE suppresses hepcidin production by interfering with the BMP/SMAD signaling pathway (Figure 5). Specifically, ERFE inhibits the induction of hepcidin by BMP5/6 and 7 [97]. This results in decreased plasma hepcidin concentrations. When hepcidin levels in the bloodstream are low, it facilitates the release of stored iron, mainly from macrophages and hepatocytes, and enhances dietary iron absorption [98].

#### 4.5.2. Growth Factors

Hepcidin, a peptide hormone produced by the liver, plays a crucial role in regulating iron absorption in the duodenum, as well as its storage and systemic distribution. Hepatocyte growth factor (HGF) and epidermal growth factor (EGF) contribute to liver regeneration by inhibiting the expression of HAMP, which is induced by iron and BMP6 [47].

#### 4.5.3. Sex Hormones

Estrogen modulates hepcidin expression through a signaling pathway dependent on G protein-coupled protein 30 (GPR30) and BMP6, suggesting that estrogen effectively reduces iron absorption in the intestine [99]. GPR30 is a membrane receptor for estrogen [100]. Furthermore, a functional estrogen response element (ERE) has been identified within the promoter region of the hepcidin gene. This finding was substantiated in hepatocytes, where 17β-estradiol was observed to reduce hepcidin levels by directly inhibiting its transcription [48]. In contrast, ferroportin mRNA transcription is significantly reduced by 17β-estradiol [101].

Hepcidin-inducing steroids (HISs) elevate hepcidin levels through a mechanism that necessitates the presence of a membrane-bound progesterone receptor, PGRMC1. PGRMC1 is identified by a single transmembrane domain and a heme-binding site [102], and it has been shown to enable rapid noncanonical progesterone signaling. Although PGRMC1 influences the function of several signal transduction pathways, the activation of hepcidin through PGRMC1 seems to rely on the activity of Src Family Kinase (SFK), while remaining independent of the PKA, PKC, or PI3 kinase signaling pathways [103]. Progesterone (50 mg daily) was administered to women undergoing in vitro fertilization, during which an increase in serum hepcidin levels was observed.

Testosterone downregulated hepcidin mRNA expression in the livers of male rats. By activating its nuclear receptor, it interferes with BMP/SMAD signaling, leading to decreased hepcidin transcription. This occurs because testosterone encourages the interaction between androgen receptors and SMAD1 and SMAD4, which decreases their binding to BMPR in men [104]. Furthermore, promotion of epidermal growth factor receptor (EGFR) signaling within the liver leads to the downregulation of hepcidin mRNA expression by BMP cell signaling [105].

### 4.6. Glucose-Mediated Signaling Pathway

Glucose regulates serum iron levels, possibly by promoting the release of hepcidin from β-cells [50]. Humans undergo rapid alterations in iron homeostasis following oral glucose ingestion, characterized by an increase in hepcidin levels and a decrease in serum iron levels within 180 min. A strong correlation was observed between serum ferritin concentration and the reduction in serum iron concentration after glucose intake. It has been suggested that the body’s iron reserves may influence the effect of glucose on serum iron levels. Additionally, the observed downregulation of hepcidin transcription in low-glucose β-cell cultures under energy or nutrient restriction may enhance iron availability [50,82]. In conditions of insulin resistance, liver iron accumulation is often observed alongside high ferritin levels and Tf saturation, which is either at the high end of normal or slightly elevated. This correlation is characterized by the upregulation of liver hepcidin mRNA expression and an increase in serum hepcidin concentrations [106,107]. In addition to their established roles in immune regulation, cytokines play a pivotal role in modulating glucose metabolism. Furthermore, the infusion of interleukin-6 in healthy individuals enhances glucose disposal by emulating hormonal activity and facilitating the oxidation of fatty acids. ALK3, also known as the BMP type 1a receptor (BMPRIA), is a BMPR I-type receptor, and BMP4 binds to ALK3 with high affinity [108]. Reduced BMPRIA signaling in β-cells diminishes the expression of essential genes involved in glucose metabolism, ultimately contributing to the development of diabetes. BMP4-BMPR1A signaling is critical for glucose-stimulated insulin secretion (GSIS) in β-cells. Thus, transgenic expression of BMP4 in β-cells and systemic dosing of BMP4 protein significantly enhance GSIS and glucose tolerance [109,110]. Moreover, insulin can directly enhance hepcidin expression in HepG2 cells by activating STAT3, but not SMAD4 expression (Figure 6) [111]. This process is partially facilitated by the extracellular signal–related kinase (ERK) pathway and occurs independently of the phosphatidylinositol 3-kinase (PI3-K) pathway [112].

### 4.7. MicroRNA

MicroRNAs (miRNAs) are short non-coding molecules that serve as crucial regulators of gene expression. MicroRNAs are small RNA sequences consisting of 19–22 nucleotides, whose main activity is the silencing of the target molecule, messenger RNA [113].

The synthesis of specific miRNAs depends on intracellular heme levels. miRNA-mediated regulation of iron metabolism can indirectly regulate HAMP expression by modifying iron uptake, storage, and release [88]. Dietary iron intake influences miRNA expression, whereas miRNAs regulate iron-related genes at the mRNA level [114]. The cytosolic concentration of iron impacts the activation of Dicer and poly(C)-binding protein 2 (PCBP2), both crucial for processing miRNA precursors [115]. Iron deficiency leads to increased pri-miRNA production [114].

miRNA 122 is a small, liver-specific, regulatory RNA. Its level depends on HFE and HJV levels [74]. Furthermore, miRNAs are stimulated by HFE or HJV and subsequently inhibit the latter, forming a negative feedback loop [88]. Inhibition of miR-122 leads to increased expression of HFE, HJV, BMPR1A, and hepcidin, which can reduce serum iron concentrations and hematopoiesis [116].

miRNA 210 expression is induced under hypoxic conditions and iron deficiency. miR-210 targets two critical molecules that play a role in maintaining iron homeostasis: TfR and iron-sulfur cluster scaffold protein (ISCU). miR-210 modulates iron homeostasis through two distinct pathways that regulate TfR expression. One pathway involves the upregulation of TfR through the indirect suppression of ISCU. The other pathway involves the downregulation of TfR via direct binding to TfR mRNA. The reduction in ISCU results in an increase in the binding activity of IRP1, while its expression level remains unchanged [117].

miR-200b plays a crucial role in the post-transcriptional regulation of FTH1 expression. FTH1 was identified as a target of miR-200b. This suggests that the increase in FTH1 expression in these cells may be triggered by low expression of miR-200b [114].

miR-320 influences the uptake of iron by cells by blocking the translation of TfR, which in turn decreases the acquisition of iron that depends on transferrin within the cells. Iron released from transferrin within endosomes is transported into the cytoplasm by DMT 1, whose expression may be suppressed by miRNA let-7d [118].

In K562 erythroleukemia cells, the overexpression of miR-let-7d leads to a decrease in both DMT1 mRNA and protein levels, which in turn diminishes the export of iron from endosomes for use by the cell. This decrease in endosomal iron export initiates an iron-deficient response, as evidenced by an increase in TfR expression, a reduction in ferritin protein levels, and a lower hemoglobin content within the cells [119].

Overexpression of miRNA 485-3p suppresses FPN expression, resulting in elevated cellular ferritin levels, which is consistent with increased cellular iron levels. In addition, miR-485-3p is upregulated during iron deficiency [120]. A decrease in miRNA 200a-3p levels may affect the translation of the iron importer, TfR1. As miRNA 194-5p levels increase, FPN protein levels decrease [121]. The miRNA 125b-5p can elevate FKN protein expression and inhibit ferroptosis [122]. miRNA 29b-3p reduces CX3CR1 levels; however, the opposite changes have been observed in the protein expression of fractalkine and CX3CR1 [121].

miRNAs play a role in ferroptosis by regulating ferroptosis-inducing genes associated with iron metabolism [123]. Specifically, FTH1 is regulated by miR19b-3p and miR-335, while FTL is controlled by miR-133 [124]. The overactivation of these miRNAs leads to an increased labile iron pool and heightened ROS production. Additionally, TfR is inhibited by miR-7-5p, and transferrin is regulated by miR-545 [125]. The downregulation of these miRNAs results in increased iron uptake into cells. Furthermore, the FPN iron exporter is regulated by miR-20 and miR-124, which can inhibit iron efflux, causing iron accumulation [126]. Recent research has shown that physical activity can alter miRNA profiles and prevent oxidative stress and ferroptosis caused by iron. Exercise influences iron-regulating miRNAs such as miR-133, miR-124, and miR-29b, which appear to offer protection to the heart [127].

### 4.8. Diet-Related Chronic Diseases and the Regulation of Iron Metabolism

Iron metabolism is related to the metabolic functions [128,129]. Metabolic syndrome (MetS) is associated with high serum ferritin levels, reflecting increased iron stores, elevated serum iron concentrations, and dysregulation of iron homeostasis. This imbalance contributes to insulin resistance and elevates the risk of type 2 diabetes (T2D) [130,131,132]. Excess iron decreases insulin receptor activity, leading to insulin resistance and reduced carbohydrate utilization in the liver and muscles [133,134]. Iron perturbation also contributes to the TNF-α and IL-6 and proinflammatory adipokines secretions of adipose tissue and the liver, and increased lipolysis [135]. Increased levels of TNF-α and IL-6 lead to a rise in HAMP expression and the secretion of hepcidin, which is followed by the reduced expression of FPN, leading to iron accumulation in the hepatocytes and macrophages [135,136]. Systemic and hepatic inflammation are both implicated in the increased production of hepcidin via the inflammatory regulatory pathway (JAK/STAT and NFκB) [137]. Although MetS is usually accompanied by iron overload, severe forms of obesity show iron deficiency [138]. The reduced serum iron levels in these patients are due to increased hepcidin concentrations, which are attributed to obesity-related inflammation [139].

Iron accumulation in patients with MetS triggers oxidative stress. Reactive oxygen and nitrogen species are harmful to macromolecules (DNA, lipids, and proteins). As the release of ROS and RNS increases, more damage occurs to these macromolecules. ROS enhances lipid peroxidation, generating lipid peroxides, which are the driving mechanism of ferroptosis, with decreasing oxidative stress protection and increasing apoptosis [140,141]. Therefore, obesity combined with chronic inflammation can cause serious organ deterioration via enhancing iron accumulation, oxidative stress, and ferroptosis [137].

Obesity is directly associated with the onset of conditions such as T2D, insulin resistance, non-alcoholic fatty liver disease (NAFLD), hypertension, and atherosclerosis. These metabolic disorders are affected by alterations in iron homeostasis [142].

Tissue iron overload due to increased hepcidin levels increases the risk of cardiovascular diseases such as coronary atherosclerotic heart disease and cardiomyopathy. The key mechanism is ferroptosis, which is mediated by oxidative stress related to iron accumulation. The nuclear factor erythroid 2-related factor 2 (Nrf2)/Keap-1 transcription factor system, which plays a role in controlling iron metabolism, is affected by oxidative stress [143]. Nrf2 binds to the antioxidant response element (ARE) on various genes, such as FTH, FTL, and FPN [144]. Nrf2 also regulates hepcidin through various signaling pathways. It may bind to the HAMP promoter or indirectly regulate inflammatory pathways and BMP/SMAD pathways, thereby inhibiting hepcidin production and preventing iron accumulation and ferroptosis in cardiomyocytes [145,146]. Cardiomyocytes uptake NTBI through voltage-dependent L-type calcium channels and zinc transporters. Ferrous iron accumulation in cardiomyocytes activates ferroptosis by decreasing glutathione peroxidase 4 (GPx4) activity and increasing the expression of NCOA4, causing the degradation of FtH and increasing the ferrous iron content of the labile iron pool [147,148]. The latter process further promotes ROS production and, therefore, lipid peroxidation and ferroptosis [149,150].

High serum iron levels, particularly non-transferrin-bound iron, contribute to atherosclerosis development by generating ROS and promoting monocyte aggregation. ROS increase LDL oxidation. The ox-LDL is engulfed by macrophages, leading to foam cell formation [151]. Moreover, excessive iron inhibits GPx, thereby facilitating ferroptosis in foam cells and advancing atherosclerosis [152]. Iron promotes endothelial dysfunction by increasing oxidative stress and ferroptosis and contributes to the adhesion of immune cells and fats to the walls of blood vessels [153].

Obesity is the primary risk factor for the onset of T2D. In this chronic disease, dysregulation of iron metabolism is implicated in the deterioration of diabetic conditions. In T2D, high serum ferritin and elevated serum iron levels have been observed [154]. Excess iron in the body’s iron stores also affects pancreatic β-cells, causing oxidative stress and damage, which leads to ferroptosis and cell death. Iron deposition in the pancreas also reduces the synthesis and secretion of insulin by β-cells. T2D can be exacerbated by high heme iron intake [155]. Hyperglycemia enhances the activity of heme oxygenase-1, causing the ferrous iron released from heme in pancreatic cells to further increase ROS production, leading to mitochondrial damage due to the inhibition of GPx4 activity [156]. Alterations in iron metabolism may also contribute to the development of insulin resistance. At the same time, insulin resistance suppresses hepcidin expression due to decreased insulin receptor activity, which can lead to further iron accumulation and damage. However, the serum hepcidin levels of T2D patients vary with the inflammation state, the degree of obesity, and insulin sensitivity [157,158].

Iron supplementation is not recommended for patients with chronic diseases associated with iron overload, such as NAFLD and T2D. These conditions can be ameliorated through dietary and pharmacological iron restriction [159]. In atherosclerosis, both iron overload and deficiency contribute to disease progression. The patient’s body iron status should be determined and considered before iron repletion or restriction [160]. A summary of iron metabolism, regulatory pathways, and diseases is presented in Figure 7.

### 4.9. Role of Iron Metabolism in Endometrium Receptivity and Implantation

Pregnancy is a special circumstance that requires fine-tuning of iron homeostasis [161]. Iron metabolism in the endometrium plays a role in endometrial receptivity and embryo implantation. Research has demonstrated that maternal iron levels significantly influence fertility by affecting the development of endometrial receptivity [162]. As a result, a lack of iron might reduce the chances of becoming pregnant and lead to infertility [163,164]. As the embryo receives nutrients, including iron, through the uterus, both iron deficiency and iron overload can affect implantation [12]. Endometrium regulates iron delivery towards the embryo through the FPN iron exporter, by releasing heme/hemoglobin or using extracellular vesicles containing iron [165]. Hepcidin suppresses iron release by causing the internalization of FPN, resulting in its low levels during the second and third trimesters of pregnancy. Nonetheless, the mechanisms that regulate hepcidin have not been thoroughly investigated. The placenta may release a specific regulatory molecule for this purpose [166]. A defect in hepcidin regulation can have serious consequences for embryo implantation and development by decreasing the availability of iron. Iron-deficiency anemia is highly prevalent during pregnancy. Increasing iron intake through supplements and a diet rich in iron can boost iron levels and improve the production of hemoglobin [167]. Elevated serum iron levels promote hepcidin synthesis, which decreases iron absorption and the release of iron to the fetus. Thus, hepcidin levels can be used as markers of responsiveness to iron therapy [168]. Using this method, excess iron supplementation can be avoided to minimize the harmful effects of oxidative stress.

In addition to hepcidin, fractalkine is another regulatory molecule that plays a role in controlling iron storage in the endometrium. This distinctive chemokine engages the fractalkine receptor (CX3CR1) to activate the inflammatory pathways NFκB, MAPK, PLC, and Nrf2 [169]. The connection between the actions of hepcidin and fractalkine is mediated by FPN [170], whose mRNA expression is regulated by the Nrf2 signaling pathway. Fractalkine exacerbates the adverse effects of iron deficiency in endometrial cells. It facilitates the redistribution of iron and modifies the cellular utilization of iron, potentially contributing to endometrial receptivity and/or ensuring iron delivery to the embryo during the early stages of pregnancy when hepcidin levels are still within the normal range [171].

### 4.10. Regulation of Iron Metabolism in Neurodegenerative Diseases

Iron is an essential element for brain function (neurotransmitter synthesis, myelination, energy production, neuronal development, and synaptogenesis). Therefore, both iron deficiency and iron overload can lead to neuronal loss [172]. Alterations in brain iron homeostasis are the driving cause of neurodegenerative diseases. However, it remains unclear whether iron accumulation in neurodegenerative diseases, such as Parkinson’s disease (PD) and Alzheimer’s disease (AD), is the cause or consequence of the developmental process [173,174]. Iron deposition is associated with α-synuclein protein aggregates in the substantia nigra in PD, whereas in AD, iron deposition appears to be linked to β-amyloid plaques [174,175].

Disruption of iron metabolism is enhanced by inflammation, infection, and brain injury [14]. Dietary iron intake, especially non-heme iron with manganese, has been implicated in the risk of Parkinson’s disease [176]. Moreover, high non-heme iron intake, either dietary or supplemental, and low vitamin C consumption increase the risk of Parkinson’s disease by 30% [177]. Iron accumulation can generate free radicals, leading to oxidative stress, the creation of lipid peroxides, and ferroptosis [178]. Parallel to the development of ferroptosis, glutathione depletion and decreased GPx activity have evolved [179,180]. Oxidative stress can lead to neuroinflammation and microglial activation, which further triggers ROS production [180,181].

Microglia regulate iron transport in the brain. These immune cells in the brain can absorb excess iron, which triggers the NFκB signaling pathway and results in the production of pro-inflammatory cytokines like IL-6, TNF-κα, and IL-1β. Moreover, microglia can release iron into the neuron [11]. Neuronal iron overload is enhanced by the downregulation of FPN and the upregulation of DMT-1 and TfR1, triggering ferroptosis in the neurons [182,183,184]. Astrocytes also contribute to neuronal ferroptosis by releasing iron and hepcidin [185].

It has been revealed that serum levels of hepcidin are higher in patients with PD or AD and are related to the severity of the disease [186,187,188]. Higher hepcidin levels are associated with lower serum iron levels [188]. These factors can serve as biomarkers for neurodegenerative diseases.

Hepcidin has two forms in the brain. The first form originates in the liver and reaches the brain via circulation and the blood–brain barrier (BBB). The second form is released by neurons and glial cells [180,189]. Inflammation causes a 40-fold increase in brain hepcidin levels. IL-6 elevation activates the JAK/STAT3 pathway, contributing to increased HAMP transcription [189]. Hepcidin released by astrocytes and microglia downregulates neuronal FPN, leading to iron accumulation [11,190]. Upregulation of BM6 in neurodegeneration may contribute to elevated hepcidin synthesis through the BMP/SMAD signaling pathway [189].

Brain hepcidin is also regulated by the fractalkine/CX3CR1 axis. Fractalkine is expressed by neurons, while CX3CR1 is expressed by microglia. Fractalkine is present in two forms: a membrane-bound form that maintains the resting state of microglia by interacting with CX3CR1 and a soluble form that is released from the neuronal membrane and activates microglia [191]. Soluble fractalkine increases hepcidin secretion of microglial cells by the internalization of CX3CR1 and promotes iron accumulation in the neurons by the internalization of neuronal FPN [11]. Fractalkine signaling is involved in the pathogenesis of neurodegeneration. However, it seems that fractalkine exerts distinct outcomes in neuroinflammation and neurodegeneration [192].

## 5. Conclusions

Iron is an essential micronutrient required for several regulatory functions in the human body. As shown in this review, it plays a role in several regulatory pathways. Understanding this pathway offers deeper insights into the mechanisms by which cells respond to fluctuations in iron demand and/or availability, thereby regulating cellular iron metabolism. In the inflammation-mediated signaling pathway, increased HAMP transcription decreases iron absorption and inhibits the release of stored iron. The iron-mediated signaling pathway can be observed in two cases: reduced iron absorption and inhibition of the release of stored iron. First, the SMAD pathway is activated via BMP ligands, and HAMP transcription is increased in cells. Second, HFE interacts with TfR2, and the HFE/TfR2 complex activates hepcidin transcription via the MAPK signaling pathway. Iron release occurs when these pathways are inhibited by SMAD 6/7 or sHJV. Hypoxia causes specific conditions. The hypoxia-mediated signaling pathway inhibits HAMP transcription, leading to iron release. In the endocrine signal-mediated signaling pathway, factors such as HGF, EGF, estrogen, testosterone, and erythroferrone suppress HAMP expression. Conversely, insulin enhances hepcidin synthesis in liver cells by activating STAT3 via the JAK/STAT signaling pathway, potentially leading to iron deficiency. miRNAs represent a distinct category of small ncRNAs, and various miRNAs potentially influence iron metabolism by regulating the genes involved in these processes. Consequently, they indirectly modulate HAMP expression by affecting iron uptake, storage, and release.

Ongoing research on iron is essential for the continued elucidation of unresolved aspects, as the pathways involved in iron regulation are not yet fully understood. A relevant example of this is the interaction between glucose and iron, as well as the process of ferroptosis, both of which still contain numerous unresolved components.

## Figures and Tables

**Figure 1 nutrients-18-00109-f001:**
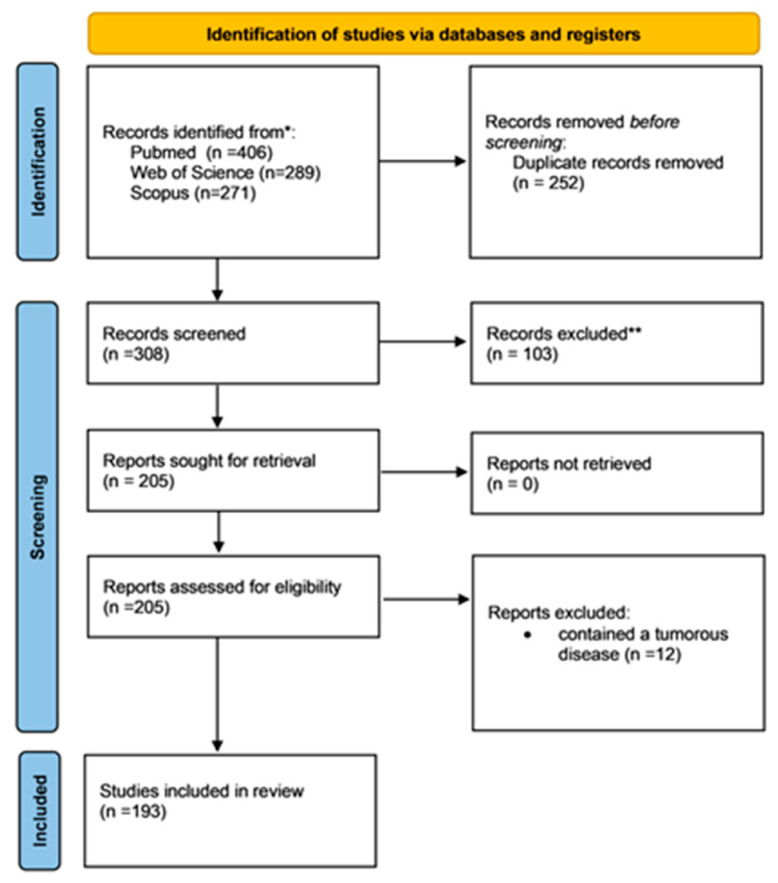
PRISMA Flowchart * consider, if feasible to do so, reporting the number of records identified from each database or register searched (rather than the total number across all databases/registers). **: If automation tools were used, indicate how many records were excluded by a human and how many were excluded by automation tools.

**Figure 2 nutrients-18-00109-f002:**
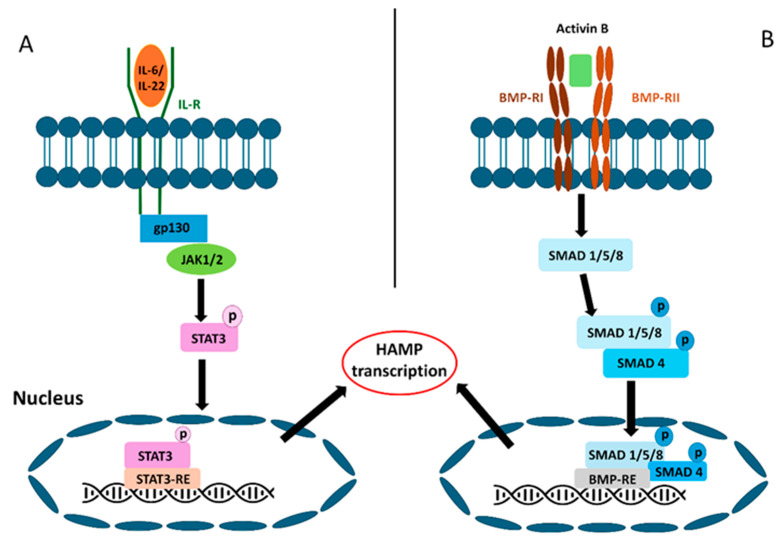
Inflammatory stimuli can induce the JAK/STAT and SMAD signaling pathways. (**A**) Inflammatory cytokines, such as IL-6 and IL-22, induce hepcidin synthesis via the JAK/STAT3 pathway. Interleukins attach to the gp130 protein receptor complex. This induces the phosphorylation of the transcription factor STAT3. JAK1/2 tyrosine kinase plays a crucial role in this process. (**B**) Another inflammatory pathway may be activated by the activin B. It can also induce synthesis through the SMAD pathway. This interaction induces the phosphorylation of SMADs, which are activated by receptors. The phosphorylated SMAD1/5/8 complex subsequently binds to SMAD4 and translocates to the nucleus, where it regulates HAMP transcription.

**Figure 3 nutrients-18-00109-f003:**
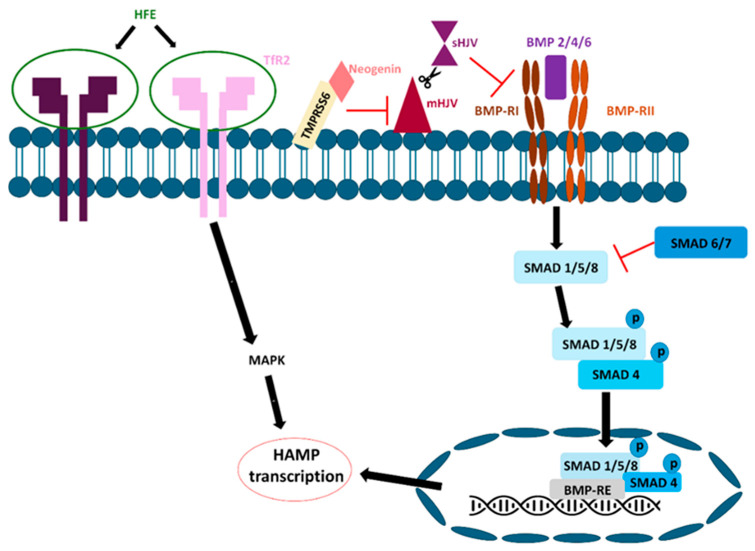
Iron-mediated signaling pathway. Hepatic cellular iron can increase the expression of BMP 2/4/6 and interact with BMPR-I and BMPR-II. This process activates the SMAD pathway, which involves the phosphorylation of SMAD1/5/8. The phosphorylated SMAD1/5/8 complex subsequently binds to SMAD4 and translocates to the nucleus, where it regulates HAMP transcription. SMAD 6/7 inhibits SMAD 1/5/8 phosphorylation. HFE interacts with TfR2 to form an HFE/TfR2 complex. The HFE/TfR2 complex activates hepcidin transcription via the MAPK signaling pathway. Neogenin subsequently activates TMPRSS6. This binding activates the proteolytic cleavage of mHJV located on the cell surface. sHJV inhibits BMP-induced hepcidin expression.

**Figure 4 nutrients-18-00109-f004:**
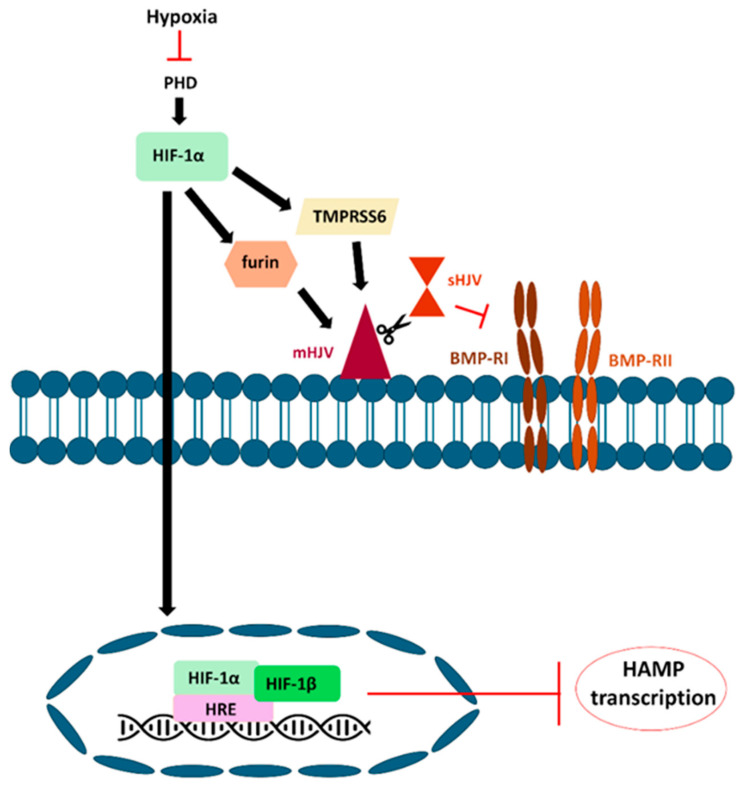
Hypoxia negatively regulates HAMP transcription. Under hypoxic conditions, PHD is inhibited, allowing for the stabilization of HIF-1. There is a direct reduction in the expression level of HAMP mRNA. Furin and TMPRSS6 can be regulated by hypoxia via HIF-1. Additionally, HIF-1α enhances the transcription of TMPRSS6, which interacts with mHJV. After cleavage, sHJV inhibits BMP-induced hepcidin expression.

**Figure 5 nutrients-18-00109-f005:**
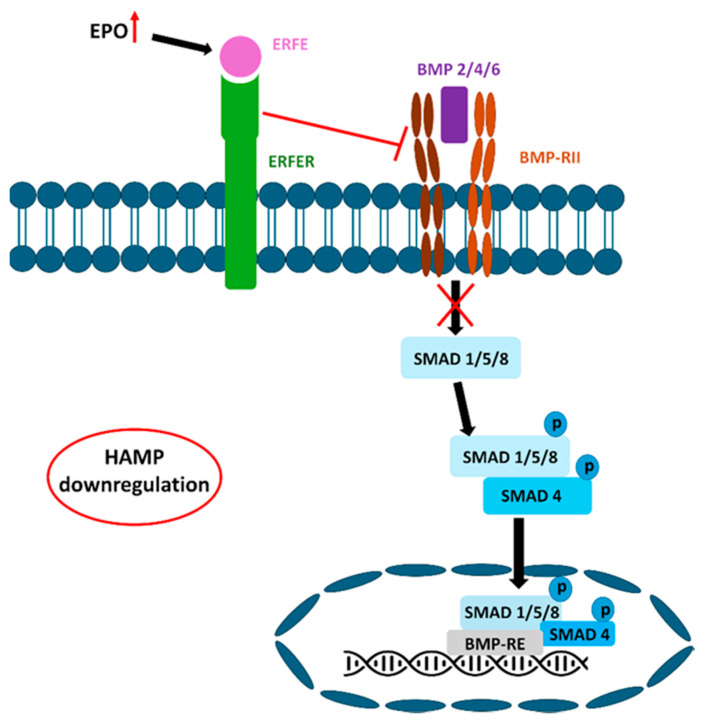
Erythroferrone inhibits the BMP/SMAD signaling pathway. If EPO production increases, ERFE synthesis also increases. Erythroferrone can inhibit BMP-RI-II and block the binding of the BMP ligand to the receptor, leading to the direct downregulation of HAMP expression.

**Figure 6 nutrients-18-00109-f006:**
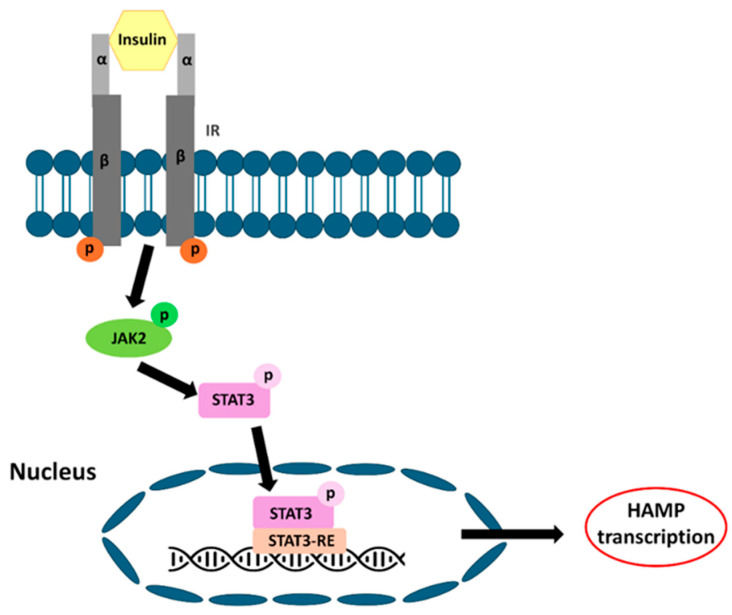
Insulin activates STAT3 via the JAK/STAT signaling pathway. The insulin receptor is a heterotetrameric membrane glycoprotein consisting of two α- and two β-subunits. ATP binding induces receptor autophosphorylation, which subsequently facilitates the receptor’s kinase activity towards intracellular protein substrates. Thus, the JAK/STAT signaling pathway is activated, which regulates HAMP transcription in the nucleus.

**Figure 7 nutrients-18-00109-f007:**
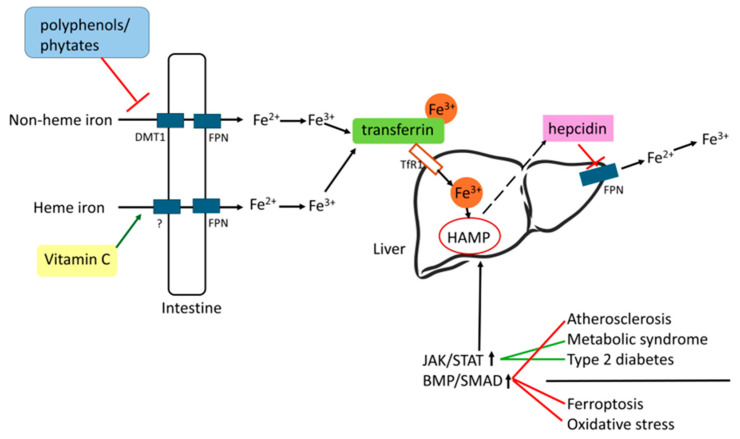
Summary of iron metabolism, regulatory pathways, and diseases. The figure shows the absorption mechanisms of non-heme and heme iron within the intestinal tract. The co-administration of vitamin C can enhance iron absorption, whereas certain compounds, such as polyphenols, may inhibit this process. Once absorbed, iron binds to transferrin and is transported to the liver via the TfR1 receptor, where it modulates HAMP transcription within the nucleus. The resultant hepcidin subsequently inhibits the liver’s FPN receptor, thereby preventing iron efflux.

**Table 1 nutrients-18-00109-t001:** The reliance of primary regulatory pathways on iron deficiency, iron excess, and variations in HAMP levels.

	Iron Deficiency	Iron Overload	HAMP Level
JAK/STAT pathway	↓	↑	↑
SMAD pathway	↓	↑	↑
BMP/SMAD pathway	↓	↑	↑
MAPK pathway	↓	↑	↑
Hypoxia-mediated signaling pathway	↑	↓	↓
EPO-mediated signaling pathway	↑	↓	↓
Glucose-mediated signaling pathway	↓	↑	↑
C/EBP signaling pathway	↓	↑	↑

## Data Availability

No new data were created or analyzed in this study.

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
