# Peer review of "Nutrients2026, 18(1), 109;https://doi.org/10.3390/nu18010109"

_nutrients, 2025, doi:10.3390/nu18010109_

Round 1

Reviewer 1 Report

Comments and Suggestions for Authors

Authors presents a very complete review on iron metabolism, that it could give an interesting starting point to discuss and future publications but it lacks of aproper discussion.

  • Search methods should be reported
  • Probably inflammation possesses a pivotal role, so it should be better considered
  • role of microRNA should be exapanded, expecially on role of diet and exercise that, are probably a regulator of iron storage
  • if possible figure s should be done with high quality, they are explicative and usefull but quite simple
  • a discussion should be included analyzing future works and possible way to improve deficencies
Comments on the Quality of English Language

it needs a revision

Author Response

Answers to Reviewer 1.

Authors presents a very complete review on iron metabolism, that it could give an interesting starting point to discuss and future publications but it lacks of aproper discussion.

  • Search methods should be reported

We appreciate your comment. The manuscript has been supplemented with a Materials and Methods section.

  • Probably inflammation possesses a pivotal role, so it should be better considered

Thank you for your advice. This section has been supplemented with additional information.

  • role of microRNA should be expanded, expecially on role of diet and exercise that, are probably a regulator of iron storage

Thank you for your advice. The miRNA section has been supplemented with the missing information.

  • if possible figure s should be done with high quality, they are explicative and usefull but quite simple

We appreciate your suggestion. The figures aim to provide a simple and clear overview of the signaling molecules involved in regulation and their interactions. We have added a summary figure describing iron metabolism, signaling pathways, and iron-related disorders (Figure 7).

  • a discussion should be included analyzing future works and possible way to improve deficiencies

An overview of iron metabolism, including therapeutic strategies, has been added to the manuscript. Please see p. 3-5.

All changes have been highlighted in the manuscript.

Reviewer 2 Report

Comments and Suggestions for Authors

Thank you for the opportunity to assess this interesting article.
The manuscript provides a detailed molecular overview of iron regulatory pathways. While the mechanistic depth is commendable, the article lacks a critical perspective and does not sufficiently align with the scope of Nutrients, which emphasizes nutritional and dietary aspects.

My major comments are listed below:

The current manuscript reads more like a biochemical review than a nutrition-oriented article. It does not address dietary sources of iron (heme vs non-heme), factors influencing absorption (e.g., vitamin C, phytates, polyphenols), or the impact of dietary patterns (vegetarian, Mediterranean) on iron homeostasis. These aspects are essential for Nutrients readership.

Besides, there is no discussion of iron metabolism in the context of diet-related chronic diseases such as atherosclerosis, metabolic syndrome, and type 2 diabetes. The role of iron in oxidative stress, LDL oxidation, endothelial dysfunction, and ferroptosis should be integrated to strengthen the nutritional and clinical relevance.

Ferroptosis is briefly mentioned in relation to microRNAs and liver injury. A more comprehensive analysis of ferroptosis as a mechanism linking iron overload, lipid peroxidation, and cardiometabolic risk would significantly enhance the manuscript.

Optional Title Revision
The current title does not reflect the nutritional dimension expected in Nutrients. While optional, I strongly recommend revising the title to emphasize the intersection of iron metabolism, diet, and disease. 

Importantly, the text is a typical narrative review – it does not include a “Materials and Methods” section or any description of literature selection criteria, databases searched, keywords used, or publication time frame. Conclusions and content rely on cited sources but lack a systematic approach (e.g., PRISMA).

Literature Integration
To provide a broader and more critical perspective, the manuscript should reference additional works, such as:

Aigner E, et al. Glucose acts as a regulator of serum iron by increasing serum hepcidin concentrations. J Nutr Biochem. 2013 Jan;24(1):112-7. 

Ru, Q., Li, Y., Chen, L. et al. Iron homeostasis and ferroptosis in human diseases: mechanisms and therapeutic prospects. Sig Transduct Target Ther 9, 271 (2024).

Sackmann A, et al. New insights into the human body iron metabolism analyzed by a Petri net based approach. Biosystems. 2009 Apr;96(1):104-13. 

Minor Comments
Please, include practical implications for dietary recommendations and supplementation strategies.
Consider adding a section on the potential risks associated with excessive iron intake from fortified foods or supplements.
Consider hemodialysis patients as a very atypical group.
Consider summarizing key pathways in a table that links molecular mechanisms to nutritional factors.

Add a Summary Figure
I strongly suggest adding a comprehensive schematic figure that summarizes the key aspects relevant to Nutrients:
Dietary sources and modulators of iron absorption (vitamin C, phytates, polyphenols).
Regulatory pathways (hepcidin, BMP/SMAD, JAK/STAT).
Links to chronic diseases (atherosclerosis, diabetes, metabolic syndrome).
Ferroptosis and oxidative stress as emerging mechanisms.  Such a complex figure would provide readers with an integrated view, thereby enhancing the manuscript's value.

Author Response

Answers to Reviewer 2.

Thank you for the opportunity to assess this interesting article.
The manuscript provides a detailed molecular overview of iron regulatory pathways. While the mechanistic depth is commendable, the article lacks a critical perspective and does not sufficiently align with the scope of Nutrients, which emphasizes nutritional and dietary aspects.

My major comments are listed below:

The current manuscript reads more like a biochemical review than a nutrition-oriented article. It does not address dietary sources of iron (heme vs non-heme), factors influencing absorption (e.g., vitamin C, phytates, polyphenols), or the impact of dietary patterns (vegetarian, Mediterranean) on iron homeostasis. These aspects are essential for Nutrients readership.

We appreciate your comment. The manuscript has been supplemented with these aspects. An overview has been written discussing the different sources of iron, absorption-influencing factors, and different diets. Please see p. 3-5.

Besides, there is no discussion of iron metabolism in the context of diet-related chronic diseases such as atherosclerosis, metabolic syndrome, and type 2 diabetes. The role of iron in oxidative stress, LDL oxidation, endothelial dysfunction, and ferroptosis should be integrated to strengthen the nutritional and clinical relevance.

We appreciate your suggestion. The manuscript has been supplemented with the relationship between iron metabolism and its regulation in the context of diet-related chronic diseases, oxidative stress, and ferroptosis. Please see Sections 4.8, 4.9, and 4.10 on pages 15-18.

Ferroptosis is briefly mentioned in relation to microRNAs and liver injury. A more comprehensive analysis of ferroptosis as a mechanism linking iron overload, lipid peroxidation, and cardiometabolic risk would significantly enhance the manuscript.

Thank you for your advice. Both ferroptosis and oxidative stress have been discussed in the manuscript. Please see Sections 4.8, 4.9, and 4.10 on pages 15-18.

Optional Title Revision
The current title does not reflect the nutritional dimension expected in Nutrients. While optional, I strongly recommend revising the title to emphasize the intersection of iron metabolism, diet, and disease. 

Thank you for your advice. Even after making the necessary modifications, our main focus remained on describing the regulation of iron management; therefore, we did not change the title.

Importantly, the text is a typical narrative review – it does not include a “Materials and Methods” section or any description of literature selection criteria, databases searched, keywords used, or publication time frame. Conclusions and content rely on cited sources but lack a systematic approach (e.g., PRISMA).

We appreciate your suggestion. A Materials and Methods section has been added to the manuscript according to PRISMA guidelines.

Literature Integration
To provide a broader and more critical perspective, the manuscript should reference additional works, such as:

Aigner E, et al. Glucose acts as a regulator of serum iron by increasing serum hepcidin concentrations. J Nutr Biochem. 2013 Jan;24(1):112-7. 

Ru, Q., Li, Y., Chen, L. et al. Iron homeostasis and ferroptosis in human diseases: mechanisms and therapeutic prospects. Sig Transduct Target Ther 9, 271 (2024).

Sackmann A, et al. New insights into the human body iron metabolism analyzed by a Petri net based approach. Biosystems. 2009 Apr;96(1):104-13. 

Thank you for your advice. The reference list has been supplemented with these publications.

Minor Comments
Please, include practical implications for dietary recommendations and supplementation strategies.
Consider adding a section on the potential risks associated with excessive iron intake from fortified foods or supplements.
Consider hemodialysis patients as a very atypical group.
Consider summarizing key pathways in a table that links molecular mechanisms to nutritional factors.

Thank you for your valuable comments. The missing information has been added to Section 3, p. 3-5. A table summarizing the key regulatory pathways and their activity states is presented in Section 4.1.

Add a Summary Figure
I strongly suggest adding a comprehensive schematic figure that summarizes the key aspects relevant to Nutrients:
Dietary sources and modulators of iron absorption (vitamin C, phytates, polyphenols).
Regulatory pathways (hepcidin, BMP/SMAD, JAK/STAT).
Links to chronic diseases (atherosclerosis, diabetes, metabolic syndrome).
Ferroptosis and oxidative stress as emerging mechanisms.  Such a complex figure would provide readers with an integrated view, thereby enhancing the manuscript's value.

Thank you for your advice. A summary figure (Figure 7) has been added to the manuscript on page 18.

All changes were highlighted in the manuscript.

Reviewer 3 Report

Comments and Suggestions for Authors

REVIEW REPORT (nutrients-4061544)

Iron, the essential micronutrient: a comprehensive review of regulatory pathways of iron metabolism.

Title: OK

Abstract:

- Although there is ample space for the inclusion of more information, I believe it is not necessary to include it due to the type of study. However, I would only indicate the need to add a direct mention of the translational potential of HAMP/hepcidin regulatory pathways, such as anemia of inflammation, iron overload.

Introduction:

- Needs to be better organized, as there are sections that appear unconnected or unexpectedly. The authors could revise the organization starting with iron and systemic homeostasis, followed by Hepcidin/HAMP as a central regulator and regulatory pathways (including BMP/SMAD, JAK-STAT, MAPK).

- It is necessary to explicitly justify the inclusion of examples such as Parkinson's disease or embryonic implantation in direct relation to hepcidin dysfunction.

- Include a final paragraph with the objective of the review.

Regulatory Pathways: Insert, at the beginning of section 2, an integrating paragraph explaining how the pathways converge for the regulation of HAMP/hepcidin.

Inflammation-mediated signaling pathway:

- Mechanisms described as fractalkine and microglia need further explanation, as they have systemic relevance or are contextual. Please review.

- The authors should clarify the heterogeneity of the effect of TNF-α on HAMP/hepcidin and also delineate the cellular versus hepatic context.

Hypoxia-mediated signaling pathway: The authors should synthesize a molecular description of HIF and focus on the final effect on hepcidin/HAMP.

Endocrine signals-mediated signaling pathway: Just make it clear in the writing that sex hormones, growth factors, and erythropoiesis should not be treated at the same level.

Glucose-mediated signaling pathway: The authors should just make it clear that it is in vitro experimental evidence versus clinical evidence.

Conclusion: What are the main regulatory convergences of HAMP? Are there any unresolved gaps? Please review these points.

Author Response

Answers to Reviewer 3.

Iron, the essential micronutrient: a comprehensive review of regulatory pathways of iron metabolism.

Title: OK

Abstract:

- Although there is ample space for the inclusion of more information, I believe it is not necessary to include it due to the type of study. However, I would only indicate the need to add a direct mention of the translational potential of HAMP/hepcidin regulatory pathways, such as anemia of inflammation, iron overload.

Thank you for your suggestion. The abstract has been modified accordingly.

Introduction:

- Needs to be better organized, as there are sections that appear unconnected or unexpectedly. The authors could revise the organization starting with iron and systemic homeostasis, followed by Hepcidin/HAMP as a central regulator and regulatory pathways (including BMP/SMAD, JAK-STAT, MAPK).

Thank you for your suggestions. The manuscript has been reorganized, and an overview of iron metabolism has been added to the article.

- It is necessary to explicitly justify the inclusion of examples such as Parkinson's disease or embryonic implantation in direct relation to hepcidin dysfunction.

We appreciate your suggestion. Sections 4.9 and 4.10 discuss these issues in detail.

- Include a final paragraph with the objective of the review.

Thank you for your advice. An objective has been added to the Materials and Methods section of the manuscript.

Regulatory Pathways: Insert, at the beginning of section 2, an integrating paragraph explaining how the pathways converge for the regulation of HAMP/hepcidin.

Thank you for your advice. A new Section 4.1 and a new table summarizing the regulatory pathways have been added to the manuscript.

Inflammation-mediated signaling pathway:

- Mechanisms described as fractalkine and microglia need further explanation, as they have systemic relevance or are contextual. Please review.

Thank you for your suggestions. Fractalkine-microglial interactions are explained in Section 4.10.

- The authors should clarify the heterogeneity of the effect of TNF-α on HAMP/hepcidin and also delineate the cellular versus hepatic context.

We appreciate your suggestion. This paragraph has been supplemented with additional information.

Hypoxia-mediated signaling pathway: The authors should synthesize a molecular description of HIF and focus on the final effect on hepcidin/HAMP.

Thank you for your advice. This section has been rewritten and reorganized to emphasize the major role of HIF factors in HAMP regulation.

Endocrine signals-mediated signaling pathway: Just make it clear in the writing that sex hormones, growth factors, and erythropoiesis should not be treated at the same level.

We appreciate your comment. This section has been modified according to the reviewer’s suggestions.

Glucose-mediated signaling pathway: The authors should just make it clear that it is in vitro experimental evidence versus clinical evidence.

We appreciate your suggestion. This signaling pathway has been supplemented with additional data.

Conclusion: What are the main regulatory convergences of HAMP? Are there any unresolved gaps? Please review these points.

Thank you for your advice. The conclusion has been modified accordingly.

All changes were highlighted in the revised manuscript.

Round 2

Reviewer 1 Report

Comments and Suggestions for Authors

Authors have improved enough the manuscript.

Comments on the Quality of English Language

it is fine

Reviewer 2 Report

Comments and Suggestions for Authors

The revised version is comprehensive, well-structured, and scientifically sound. It addresses the topic in depth, integrates recent literature, and provides precise figures and tables. The conclusions are consistent with the objectives and highlight relevant therapeutic implications. 
The authors have considered all my comments and suggestions. I'm satisfied. Well done.

Reviewer 3 Report

Comments and Suggestions for Authors

None